# Stretchable Strain Sensor with Small but Sufficient Adhesion to Skin

**DOI:** 10.3390/s23041774

**Published:** 2023-02-04

**Authors:** Takaaki Nishikawa, Hisaya Yamane, Naoji Matsuhisa, Norihisa Miki

**Affiliations:** 1Department of Mechanical Engineering, Keio University, Kanagawa, Yokohama 223-8522, Japan; 2Department of Electronics and Electrical Engineering, Keio University, Kanagawa, Yokohama 223-8522, Japan; 3JST PRESTO, 7 Gobancho, Chiyoda-ku, Tokyo 102-0076, Japan; 4Institute of Industrial Science, The University of Tokyo, Meguro-ku, Tokyo 153-8505, Japan

**Keywords:** flexible sensor, strain sensor, adhesion, human motion, silicone rubber, liquid metal

## Abstract

Stretchable strain sensors that use a liquid metal (eutectic gallium–indium alloy; E-GaIn) and flexible silicone rubber (Ecoflex) as the support and adhesive layers, respectively, are demonstrated. The flexibility of Ecoflex and the deformability of E-GaIn enable the sensors to be stretched by 100%. Ecoflex gel has sufficiently large adhesion force to skin, even though the adhesion force is smaller than that for commercially available adhesives. This enables the sensor to be used for non-invasive monitoring of human motion. The mechanical and electrical properties of the sensor are experimentally evaluated. The effectiveness of the proposed sensors is demonstrated by monitoring joint movements, facial expressions, and respiration.

## 1. Introduction

Strain sensors transduce mechanical deformations into electrical signals. Soft robust strain sensors have recently been developed by the incorporation of advanced nanomaterials into stretchable support materials [1,2,3,4]. To date, highly stretchable strain sensors have been fabricated using low-dimensional carbon (e.g., carbon black (CB), carbon nanotubes (CNTs), and graphene), nanowires (NWs), nanoparticles (NPs), MXenes, liquid metals, and hybrid micro/nanostructures [1,2,3,4,5,6,7,8,9,10,11,12,13,14,15,16,17,18]. Silicone-based elastomers (e.g., polydi-methylsiloxane (PDMS), Ecoflex, and Dragon Skin), rubbers (e.g., natural rubber and thermoplastic elastomers (TPEs)), and hydrogels are the most commonly used flexible support materials for stretchable strain sensors [1,2,4,6,7,8,9,11,12,13,19,20]. Depending on the combination of materials and structures, some sensors have been developed with high elasticity (ε = 800) and others with high sensitivity (GF = 2000) [13,17].

Promising applications of these stretchable strain sensors include health monitoring, human motion detection, and human-machine interfaces [5,11,12,21,22,23,24,25,26,27] in which the sensors are attached to the skin. Sensors with sufficient stretchability can measure characteristics such as muscle movement, joint movement, heartbeat, and respiration, while maintaining good contact with the skin. Maintaining adhesion between a sensor and skin is a challenging problem. Most skin-mounted strain sensors reported to date are attached to the skin using commercially available adhesives [19]. Although these provide good adhesion to the complex and highly deformable skin and enable high-quality sensing, the large force (37–62 N/m on pig skin) required to peel off such sensors may damage the skin [18,28]. Skin damage associated with peeling off of medical adhesives frequently occurs for neonates and the elderly [29]. Appropriate control of the adhesion force to ensure good contact between a sensor and the skin, and allow the sensor to be removed without causing damage, would allow expansion of the scope of applications.

In the present study, a highly stretchable strain sensor composed of a sensing layer made of liquid metal, silicone rubber, and an adhesive layer of silicone is demonstrated. The structure of the proposed sensor is shown in Figure 1. The sensor is found to exhibit good adhesion, but can be removed by applying a low peeling force. The electrical and adhesive properties of the strain sensor are characterized, and successful measurements of finger-joint movement, facial movement, and respiration are presented.

## 2. Materials and Methods

### 2.1. Material

Eutectic gallium–indium alloy (E-GaIn: Ga 75.5%/In 24.5%, ≥99.99% trace metals; Sigma-Aldrich, St. Louis, MO, USA) was used as the liquid metal to form the conductive part. The resistivity of E-GaIn is 2.9 × 10^−7^ Ω·m [30]. With a melting point of 15.5 °C, this alloy is a liquid at room temperature and has no mechanical deformation limits. E-GaIn has been reported to be non-toxic to humans [31,32]. Ecoflex^®^ 00-10 (Smooth-On Inc., Macungie, PA, USA) was used as the silicon elastomer. Ecoflex^®^ Gel (Smooth-On Inc.) was used as the adhesion layer. Ecoflex^®^ 00-10 can be stretched by up to 800% and is easily deformed due to its low Young’s modulus [33,34]. Ecoflex^®^ Gel has adhesive properties and high elasticity, which allows the flexible stretchable sensor to adhere to skin. Copper foil (0.05 mm thick, AS ONE Corp., Osaka, Japan) was used for connections between the liquid metal conductors and external measurement instruments. Epoxy resin-based adhesive (Bond Quick 5, Konishi Co., Ltd., Saitama, Japan) was used to secure the copper foil to the support material.

Stiffness is proportional to the cube of thickness. It has been reported that a 100-μm-thick elastomer film can conform to the wrinkles and fine curvature of skin due to its low stiffness [18]. The film should have high stretchability to survive strains of up to 60–65% [35,36]. In addition, its Young’s modulus must be comparable to that for human skin (E = 25–260 kPa) to ensure that the sensor does not interfere with natural skin movement [37].

The thickness and Young’s modulus for Ecoflex^®^ 00-10 thin films fabricated by spin-coating were investigated. In this study, 3.5 mL of Ecoflex^®^ 00-10, a mixture of Agents A and B in a 1:1 ratio, was dispensed onto a 76 × 52 mm glass slide and spin-coated (MS-B100, Mikasa Co., Ltd., Osaka, Japan). The spinning rate was increased for 5 s to the target value, held for 60 s, and then decreased for 5 s to zero. The film thickness was measured with a laser microscope (VK-X100, Keyence Corporation, Osaka, Japan) after a thin film of copper was deposited on the surface by vapor deposition. The thickness of the film was measured at 2 mm intervals from the center to the edge of the long axis of the glass slide. Figure 2 shows the measured dependence of the Ecoflex^®^ 00-10 film thickness (data points) on the spinning rate. The film was formed uniformly and spatial thickness differences were found to be negligible. The curve in the figure is the theoretical relationship derived by Koschwanez et al. and Emslie et al., between the spin time t [s], angular velocity ω [rad·min^−1^], and thickness h [μm] [38,39]:(1)$$h=\frac{h_{0}}{\sqrt{1+c\omega^{2}h_{0}{}^{2}t}},$$
where $h_{0}$ and c were experimentally determined to be 3000 μm and 1.38 × 10^−14^ min^2^·rad^−2^·μm^−2^·s^−1^, respectively. A 100-µm-thick film can be fabricated at 1500 rpm.

Young’s modulus for the films was determined by tensile testing. Dumbbell-shaped specimens, as shown in Figure 3a, were mounted on a tensile testing machine (MST-I HR, Shimadzu Corporation, Japan) and tensile tests were performed at a rate of 30 mm·min^−1^. The distance d between the upper and lower clamps was varied from $d_{0}=18$ mm to $d_{max}=72$ mm, and a tensile strain of 300% was applied to obtain stress–strain curves. As a control experiment, a tensile test was conducted using polydimethylsiloxane (PDMS; 10:1 mixture of main and cross-linking agents, SILPOT™ 184W/C, Dow Toray Co., Ltd., Tokyo, Japan), a commonly used support material for sensors. Young’s modulus was determined by averaging the values at 60–65% strain, and the results were averaged over three specimens fabricated under the same conditions. Young’s modulus for Ecoflex^®^ 00-10 and PDMS was determined to be 43.3 and 353 kPa, respectively (see Figure 3b). Thus, Young’s modulus for Ecoflex^®^ 00-10 was approximately 8 times lower than that for PDMS, and comparable to that for human skin (25–260 kPa). Young’s modulus for PDMS has been reported to vary with film thickness [40]. However, in the range of interest of 100 to 500 μm, Young’s modulus for Ecoflex^®^ 00-10 was experimentally found to not be affected by film thickness (Figure 3c).

### 2.2. Fabrication of Stretchable Sensor

The sensor was composed of two layers of elastomer, one of which was adhesive and the other non-adhesive (see Figure 1). Figure 4 shows the fabrication process, where both layers were produced by spin-coating. First, a glass slide was treated with Cytop™ (CTL-809M mixed with CT-solv180 in a 1:10 solution, AGC Inc., AGC Chemicals, Fukui, Japan). The first layer of non-adhesive Ecoflex 00-10, a mixture of Agents A and B at a volume ratio of 1:1 and 3.5 mL, was dispensed on the slide and spin-coated at 1500 rpm for 60 s. Degassing was performed in a vacuum and curing was conducted for 2 h at room temperature. A copper foil was then attached to the cured Ecoflex film with an epoxy resin-based adhesive and a curing time of 1 h. The conductive part of the sensor was formed by spraying E-GaIn using an airbrush through a stencil mask made of polyimide and formed by laser cutting. A volume of 3.5 mL of Ecoflex^®^ Gel, a 1:1 mixture of Agents A and B, was then spin-coated onto the first layer with the patterned conductive layer at a spinning rate of 700 rpm for 60 s. After degassing in a vacuum and curing at room temperature for 2 h, the layers were cut with a scalpel and peeled off from the glass slide for use.

### 2.3. Characterization of Stretchable Sensor

#### 2.3.1. Electrical Properties

The electrical properties of the sensor, which include electrical robustness to strain, sensitivity, linearity, hysteresis behavior, and response time, were investigated. In the experiments, the second layer of adhesive Ecoflex^®^ Gel was replaced with non-adhesive Ecoflex^®^ 00-10. The resistance of the sensor was measured with a digital multimeter (DMM6500 6 1/2, Keithley Instruments, Cleveland, OH, USA) in a 4-wire resistance measurement mode, while strain from 5% to 100% at 5% intervals was applied by the tensile testing machine. A maximum strain of 100% was used because the maximum strain expected for the wearable application is 65% [35,36]. Tensile strain was applied at a rate of 67%/min. The tests were repeated five times for each strain value. The gauge factor for the sensor was determined from this experiment. To evaluate the sensitivity of the sensor for very small strain values, strains of 0.1 to 0.01% were used. The hysteresis, drift, and response time for the sensor were then investigated by repeating the tests 1000 times with a strain of 100%. Experiments were conducted at 22 °C and 40% humidity.

#### 2.3.2. Adhesion

A peel test was performed to determine the adhesion strength of Ecoflex Gel when in contact with human skin. The inner forearm of a subject (the author, male, 23 years old) was wiped with a paper towel (Kim Towel White) containing deionized water. After the skin was dried, the sensor was applied to the subject’s forearm along the long axis of the arm and was crimped with a 1130 g roller with an adhesive area of 50 × 70 mm (see Figure 5). One minute after application, the end of the sensor was attached to a chuck connected to the load cell of the tensile tester. The load cell was moved at a rate of 30 mm·min^−1^ and the sensor was peeled off at an initial peeling angle of 90° in the longitudinal direction. Peeling tests were performed three times. As a control experiment, commercially available adhesives (commercial adhesive A: Temporary Tattoo Paper, Silhouette America Inc., Lindon, UT, USA; commercial adhesive B: Substrate-less MX (Matrix^®^) double-coated adhesive tape, Nichiei Kako Co., Osaka, Japan) were also tested.

To evaluate the reproducibility and durability of the adhesive, the effects of repetitive application and peeling off of the sensor were investigated. Application and peeling of the sensor were repeated 10 times. After peeling and before application, the surface of the Ecoflex^®^ Gel was wiped with ethanol and the subject’s inner forearm was wiped with deionized water. Experiments were conducted in a room with a temperature of 22 °C and 40% humidity.

### 2.4. Demonstration

Facial expression, respiration, and joint movement were monitored with the strain sensor attached to a measurement site (see Figure 6). During the experiments, the copper foil to interconnect the sensor and the external circuit was firmly attached to the skin to prevent artifacts due to movement. When monitoring facial expression, the subject was requested to smile. Respiration was monitored while the subject was in a relaxed state sitting in a chair. The motion of the joints of the elbow, knee, finger, wrist, and ankle was assessed by the attached sensors. The sensors were attached to joints when they were extended (the angle was 0°). For the sensors at the elbow, knee, and finger, the resistance was measured at joint angles of 45°, 90°, and the limit of the flexion angle. At the wrist and the ankle, the resistance was measured during dorsiflexion and palmarflexion. Experiments were conducted five times for each condition.

## 3. Results and Discussion

### 3.1. Mechanical and Electrical Properties of Stretchable Sensor

As shown in Figure 7a, $\Delta R/R_{0}$ varied from 9.8% to 436% as the strain increased from 5% to 100%. The change in resistance originates from the deformation of the sensor. Large deformation of the stretchable sensor leads to a large variation in resistance. However, the deformation is nonlinear; therefore, the gauge factor increased with increasing strain from 1.96 to 4.36, as shown in Figure 7b. The results are shown in Figure 7c. It can be seen that there is a significant difference between a strain of 0% and 0.01%, although there is some variation due to noise. This verified that the sensor can successfully measure a strain of 0.01%.

Stretching at a rate of 67%/min did not cause any notable delay in the response (see Figure 8a). Hysteresis of the sensor was investigated for maximum strains of 20%, 50%, and 100%. Small hysteresis was observed at a strain of 20%, while a larger hysteresis was observed at 50 and 100%, as shown in Figure 8b [41,42]. Figure 8c shows $\Delta R/R_{0}$ in a repeat test with 1000 cycles of 100% strain. Strain was applied at a rate of 67%·min^−1^. $\Delta R/R_{0}$ at 100% increased by 22% in the first 50 cycles and then by only 7% in the next 950 cycles, which is considered to be due to slight plastic deformation of the Ecoflex support material. The liquid metal showed sufficient durability as there was no fatigue due to deformation and no significant change in Ecoflex after 1000 cycles. The results verified the reliability of the sensor. A total of 10,000 cycle tests will be conducted in a future study.

### 3.2. Adhesion Tests

The results of the Ecoflex^®^ Gel peel tests are shown in Figure 9. The peel strength initially increased and then plateaued when the tester stroke was approximately 15 mm. As peeling progressed, the peel strength increased slightly due to the effect of the smaller peeling angle. The average peel strength in the plateau region around a stroke of 15 mm was used as the peel strength for the adhesive. The peel strengths for commercial adhesives A and B were 53.9 N/m and 45.3 N/m, respectively, while the peel strength for Ecoflex^®^ Gel was 3.70 N/m. This indicates that adhesion with Ecoflex^®^ Gel requires only one-tenth of the peel-off force needed for conventional adhesives. An adhesive peel test over 10 cycles was conducted to evaluate the reproducibility and durability of adhesion. It has been reported that self-adhesive sensors may lose adhesive strength after repeated wearing and peeling [8]. As shown in Figure 9b, the sensor we developed did not show any decrease in adhesive strength after 10 repetitions. We consider 10 repetitions to be sufficient, since the sensors are in direct contact with the skin, and users may not like to reuse them many times.

When the sensor is stretched by skin deformation, the elastic force of the support material acts to cause peeling. The skin-mountable sensor must have an adhesive strength that does not lead to peeling due to its own elasticity, and this can be evaluated by a comparison of the elastic force and the peeling force at 65% strain, which is the maximum strain that can occur on the skin. The elastic force *f_e_*, generated by the sensor at 65% strain is as follows:(2)$$f_{e}=E_{1}h_{1}w\times 0.65=4.22\times 10^{-2}\;N,$$
where $E_{1}$ and $h_{1}$ are Young’s modulus for Ecoflex (43.3 kPa) and the thickness of the support layer (100 μm), respectively. Here, w is the width of the sensor (15 mm). The direction of the force caused by skin strain is tangential to the skin surface. Therefore, the elastic force causes peeling at an angle of 0°. The peeling force is dependent on the peeling angle between 0° and 90°, and has a minimum at 90°. The peeling force at 0° is 10 times larger than that at 90° [43]. The peel strength at a peeling angle of 90° was determined experimentally to be 3.70 N/m, and the peeling force was found to be 5.55 × 10^−2^ N. The elastic force is smaller than the peeling force at a peeling angle of 90°, which indicates that the sensor does not peel off due to skin strain.

### 3.3. Demonstration

To further demonstrate the application of the wearable device, various human body activities and joint movements were monitored using the developed sensor. As shown in Figure 10, when the sensors were attached to the index finger, elbow, and knee, the resistance increased with joint flexing, and immediately returned to the original value when the bending angle was returned to 0°. When the flexion angles were changed to 45°, 90°, and full flexing, a larger change in resistance was observed with a larger flexing angle. As shown in Figure 10, the sensor adhered to the skin and followed the flexion and extension of the joints, and the strain was successfully obtained. The resistance change for each joint at full flexion was 0.40 for the index finger, 0.99 for the elbow, and 0.74 for the knee. Using a function obtained from the evaluation of the electrical characteristics of the sensor, the average strain for the measured interval was determined to be 17.5% for the index finger, 36.0% for the elbow, and 28.8% for the knee. The corresponding strains measured with a ruler were 24.4%, 44.4%, and 35.6%, indicating that the sensor underestimated the strain. This was because the firmly attached copper foil interfered with natural skin movement. When the sensors were used to detect wrist and ankle motion, it was necessary to follow the wrinkles that occur during dorsiflexion. As shown in Figure 11, the sensor can follow wrinkles during dorsiflexion, and detect dorsiflexion motion by the decrease in the change in resistance. In dorsiflexion, as in other joint flexions, the resistance returned to the original value as soon as the joint returned to its natural state. Monitoring human body activity requires detection of not only large strains such as those associated with joint movements, but also small strains such as those caused by changes in facial expression and stomach movements during breathing. For the detection of facial expression, the system showed a highly reproducible response to rapid changes with a short response time (see Figure 12a). In the detection of respiratory rate, it was observed that the resistance change increased when the stomach was inflated by inhalation and decreased when the stomach was deflated by exhalation. The peaks and valleys in Figure 12b indicate the state of inhalation and exhalation. The large overall fluctuations may be due to movement of the digital multimeter wires or local elongation of the connection part.

In all the experiments, the sensor could be peeled off from the skin with ease. This non-invasiveness expands the medical applications of the proposed sensor, even for neonates and the elderly.

## 4. Conclusions

A stretchable strain sensor was fabricated using E-GaIn liquid metal and flexible Ecoflex silicone rubber as the support, together with an adhesive layer. The electrical and mechanical properties, including the adhesion properties, were experimentally investigated. The sensor could be stretched by 100%, and the gauge factor changed from 1.96 to 4.36. Young’s modulus for the sensor was smaller than that for a PDMS-based sensor and was comparable to that for human skin. The adhesion force for Ecoflex Gel was smaller than that for commercially available adhesives, but was still sufficiently large for the sensor to be fixed to human skin, even at the joints. Successful monitoring of the movement of joints, facial expressions, and respiration, verified the effectiveness of the proposed sensor.

## Figures and Tables

**Figure 1 sensors-23-01774-f001:**
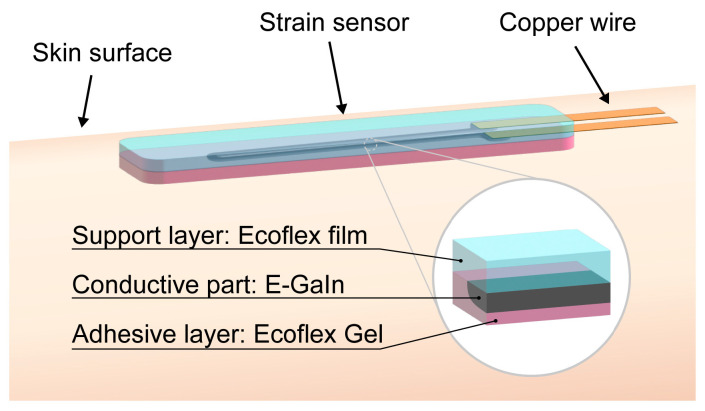
Illustration of a stretchable and detachable sensor.

**Figure 2 sensors-23-01774-f002:**
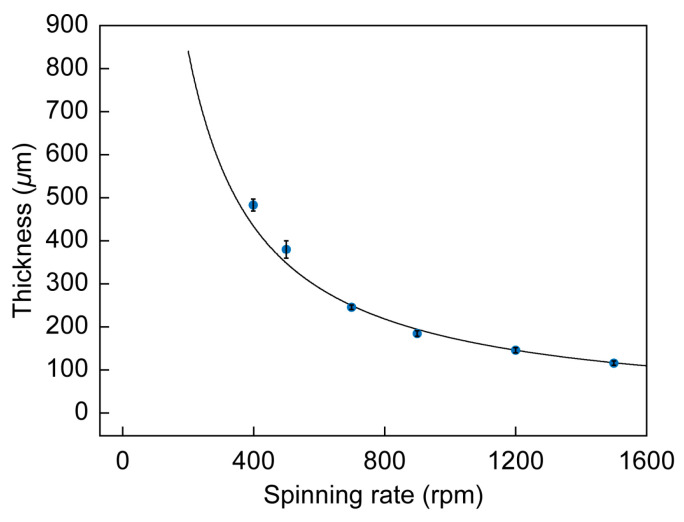
Thickness of Ecoflex^®^ 00-10 films as function of spinning rate.

**Figure 3 sensors-23-01774-f003:**
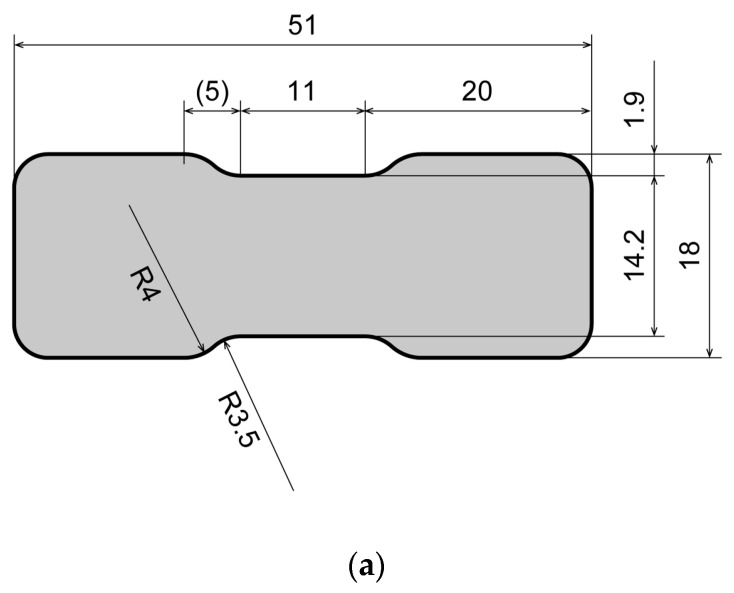
(**a**) Specimen used for tensile test (dimensions in millimeters). (**b**) Young’s modulus for Ecoflex^®^ 00-10 and PDMS. (**c**) Young’s modulus for Ecoflex^®^ 00-10 films deposited at various spinning rates.

**Figure 4 sensors-23-01774-f004:**
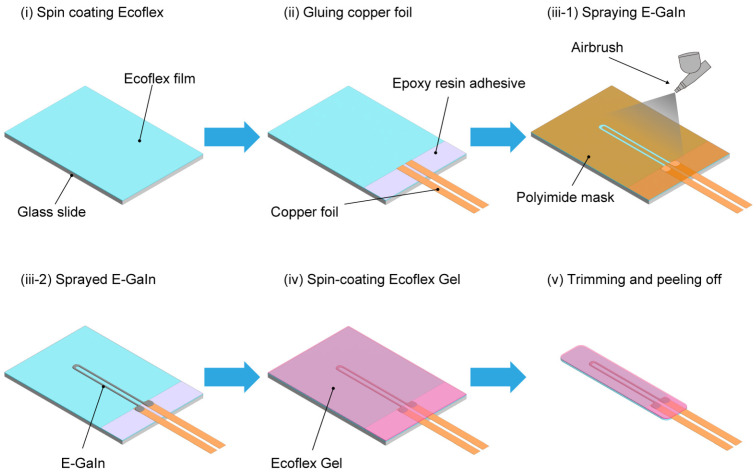
Fabrication process for stretchable strain sensor.

**Figure 5 sensors-23-01774-f005:**
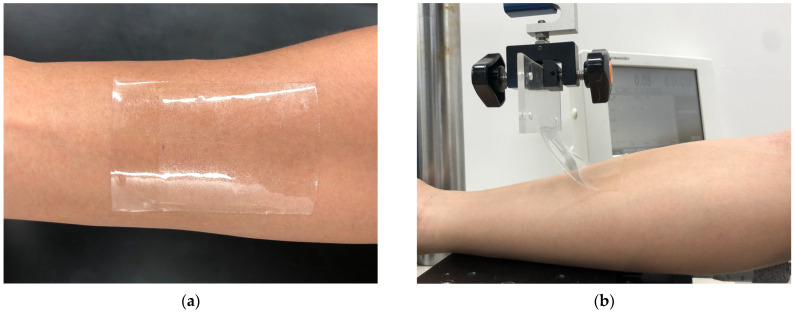
Peel tests for sensor. (**a**) Pasting point; (**b**) Photo of peel test.

**Figure 6 sensors-23-01774-f006:**
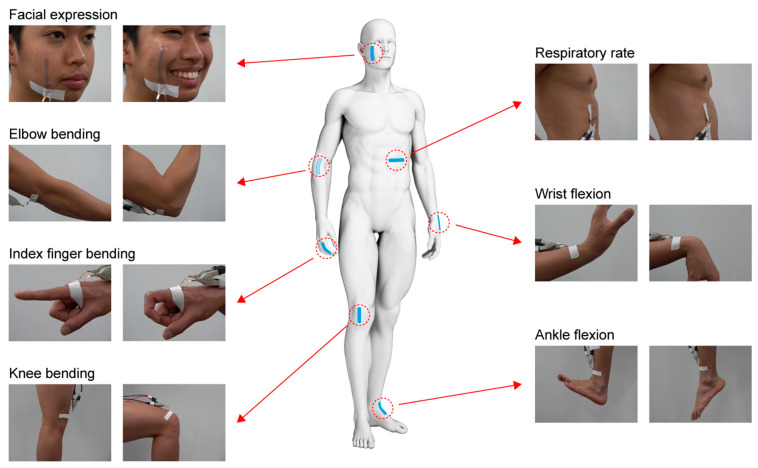
Measurement sites for human activity monitoring by stretchable strain sensor.

**Figure 7 sensors-23-01774-f007:**
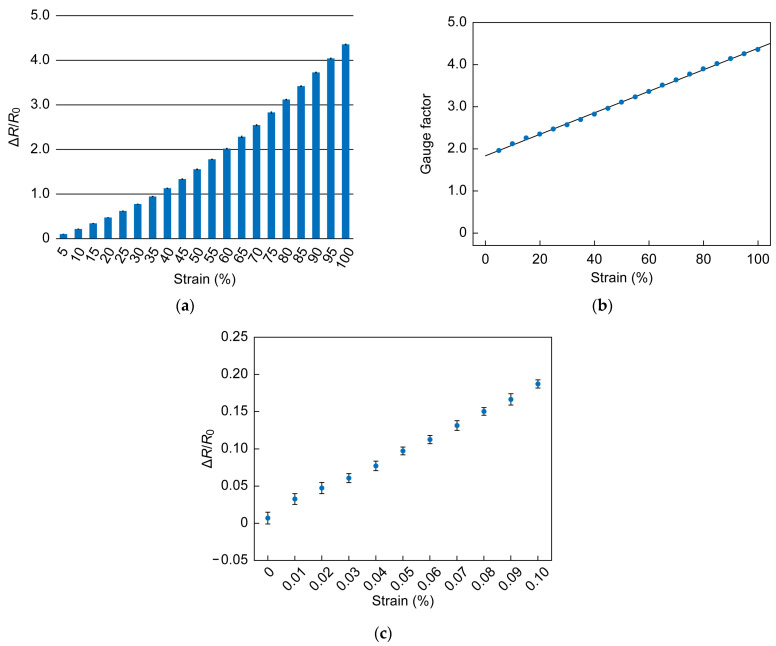
(**a**) Response of sensor to 20 different levels of strain, ranging from 5% to 100% at 5% intervals. (**b**) Gauge factor as function of strain. (**c**) Response of sensor to strain ranging from 0% to 0.1%.

**Figure 8 sensors-23-01774-f008:**
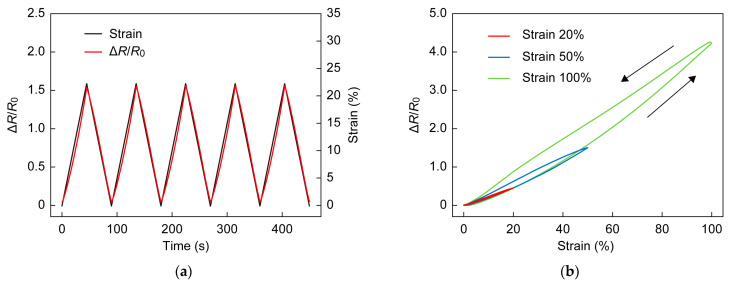
(**a**) Real-time response under application of strain to sensor. (**b**) Sensor hysteresis for 20%, 50%, and 100% stretching. (**c**) Repeat stretching tests for 1000 cycles.

**Figure 9 sensors-23-01774-f009:**
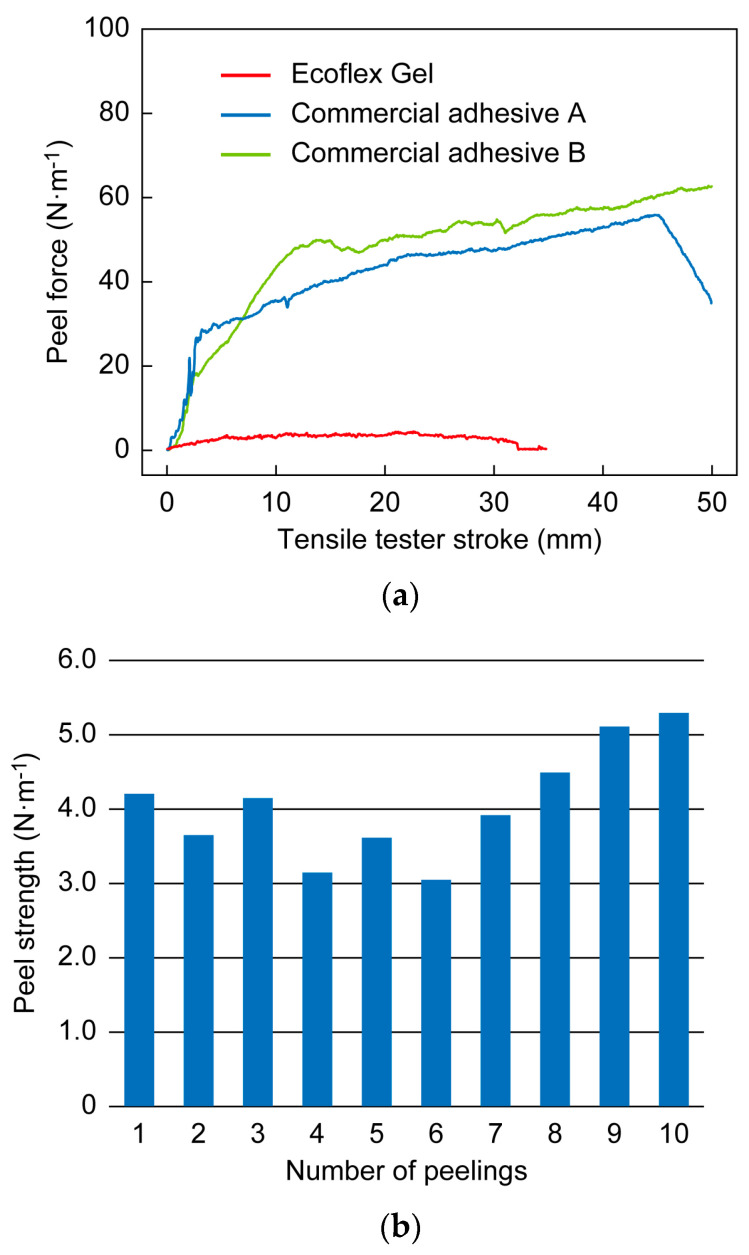
(**a**) Peel strength of Ecoflex Gel compared to commercial adhesives. (**b**) Peeling strength during 10 cycles of adhesion and peeling.

**Figure 10 sensors-23-01774-f010:**
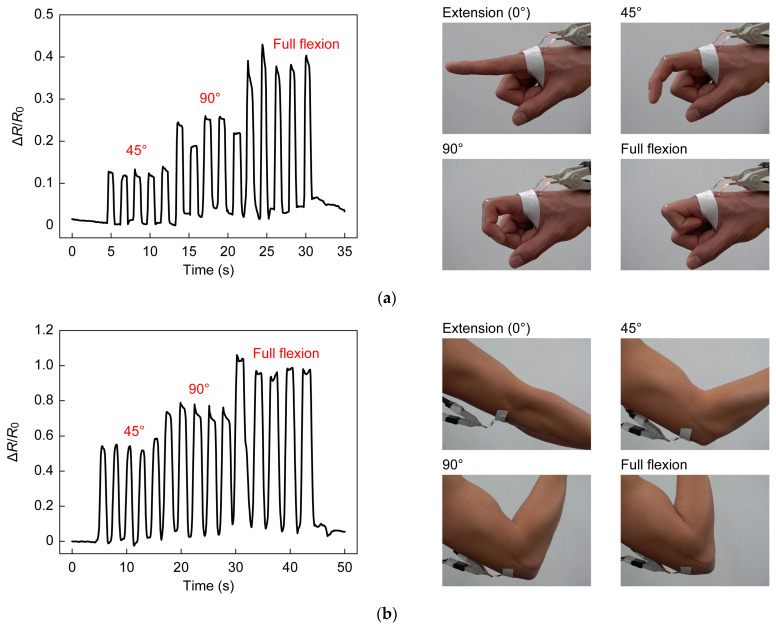
(**a**) Images of finger flexions and sensor responses; (**b**) Images of elbow flexions and sensor responses; (**c**) Images of knee flexions and sensor responses.

**Figure 11 sensors-23-01774-f011:**
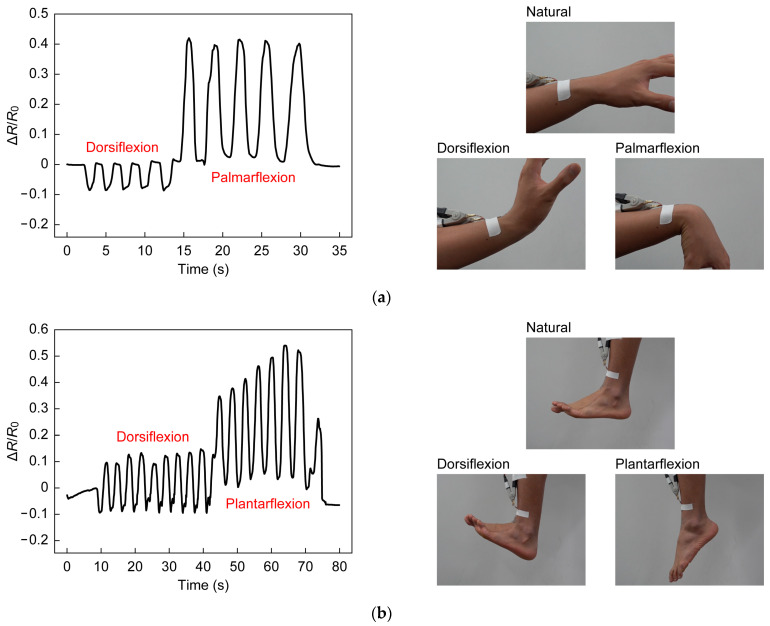
(**a**) Images of wrist flexions and sensor responses; (**b**) Images of ankle flexions and sensor responses.

**Figure 12 sensors-23-01774-f012:**
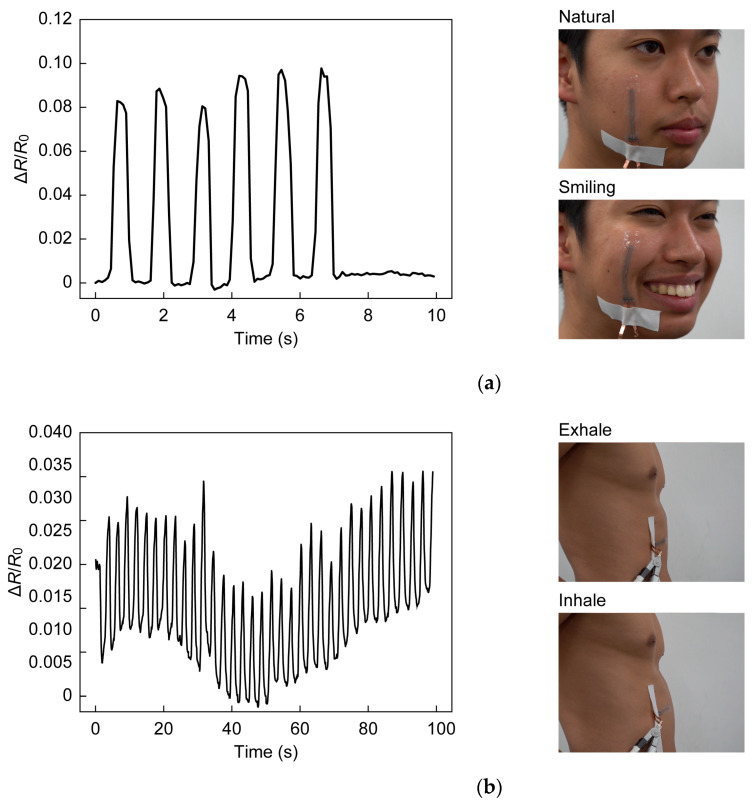
(**a**) Images of facial expressions and corresponding sensor response. (**b**) Images of respiration and corresponding sensor response.

## Data Availability

The data presented in this study are openly available in FigShare at doi at https://doi.org/10.6084/m9.figshare.21780140 (accessed on 25 December 2022).

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
