# Peer review of "Stretchable Strain Sensor with Small but Sufficient Adhesion to Skin"

_sensors, 2023, doi:10.3390/s23041774_

Round 1

Reviewer 1 Report

Review:

The article “Stretchable Strain Sensor with Small but Sufficient Adhesion 2 to Skin” is based on the stretchable strain sensors that use a liquid metal (eutectic gallium-indium alloy; E-GaIn) and flexible silicone rubber ability. The experiments are performed well but this work can not be accepted in its present form. Work can be considered for potential publications after the following major changes/clarifications:

1.       I will like to know if the authors had considered the effect of GaIn metal on humans, in case someone gets into direct contact with the liquid. Explain briefly.

2.       The authors used the word large adhesion, does it mean it covers large area or regarding the strength?

3.       The proposed sensors are reusable, and if yes then for how many times?

4.       The keywords should be reduced to six.

5.       The paper is submitted in Dec, so most probably it will be published in 2023, however paper does not have references from 2023 and only two references from 2022. I will like to suggest authors to add some more references like: https://doi.org/10.1111/cote.12581.

6.       Please refer to some articles where GaIn was used for its strain sensor applications for comparative analysis.

7.       English need to be revised like there are many sentences with obvious grammatical errors e.g., Successful demonstration, in which the sensors were used monitor the movement of the joints, facial expressions, and respiration, verified the effectiveness of the proposed sensors.

8.       What was the spinning rate to achieve 100-micron thickness?

9.       Can you kindly explain the expression c=1.38c=1.38x10^-14, and was ‘t’ constant?

10.   Any reason for the difference in PDMS and Ecoflex Young’s modulus

11.   What was the main reason behind high hysteresis at 100% stress?

12.   Mention Figure 9 (b) in the theory.

Author Response

Attached you can find the revision we have made following your precious comments.

Reviewer 2 Report

The authors proposed and fabricated a stretchable strain sensor using E-GaIn liquid metal as conductive materials and flexible Ecoflex silicone rubber as the support and adhesive layers. The electrical and mechanical properties, including the adhesion properties, were experimentally investigated. Importantly, the adhesion force of Ecoflex Gel was smaller than that of the commercially available adhesive while it was sufficiently large for the sensor to be fixed to the human skin, even at the joints. But the experiment exists some deficiencies. The data seemed insufficient.

1.      The stiffness and Young’s modulus are important to maintain good contact with the skin. As shown in Figure 2, the thickness of the Ecoflex® 00-10 films as a function of the spinning rate. And the thickness should highly affect the stiffness and the Young’s modulus. But the author demonstrated that no significant difference in the Young’s modulus of the Ecoflex® 00-10 films was observed with respect to the thickness or spinning rate used for spin-coating in Line 109-110 and in Figure 3c. Please make explain.

2.      For electrical properties of the stretchable sensor, I think the experiments are not enough. For example, in Figure 7, only testing up to 100% strain is obviously too small. And in order to show the high gauge factor and the potential application in facial expression, the detect limitation should much lower than 5% strain. Thus, the author should enlarge the testing range of strain. If the repeat test may be done for more than 10000 cycles the discuss in reliability will more credible.

3.      Please make comparison of the electrical properties of the stretchable sensor in this work to the published works, especially using Ecoflex file as support and liquid metal as conductive materials.

4.      In the design of experiment, another problem is that the author only used a single sample to investigate the electrical properties. Why not applied the samples with different thicknesses of Ecoflex film?

5.      In line 208-209, the author said that the peeling force at 65% strain is the maximum strain that can occur on the skin. Please cite the reference.

Author Response

(The authors gave the same response as above.)

Round 2

Reviewer 1 Report

Dear Authors,

Thank you for making changes as per suggestion.

I would like to inform that according to me, the paper can be accepted in the paper in the present form.

Good luck.

Reviewer 2 Report

In this revised manuscript, the authors answered all my questions and made considerable revision. Now I have no more questions and suggest to accept this manuscript.